# TCF7L1 Controls the Differentiation of Tuft Cells in Mouse Small Intestine

**DOI:** 10.3390/cells12111452

**Published:** 2023-05-23

**Authors:** Valeriya V. Zinina, Melanie Sauer, Lira Nigmatullina, Nastasja Kreim, Natalia Soshnikova

**Affiliations:** 1Institute for Molecular Medicine, University Medical Center of the Johannes Gutenberg-University, 55131 Mainz, Germany; vazinina@uni-mainz.de (V.V.Z.); melanie.sauer@student.uni-tuebingen.de (M.S.); 2Institute of Molecular Biology gGmbH, 55128 Mainz, Germanyn.kreim@imb-mainz.de (N.K.); 3Institute for Molecular Medicine and Research Center for Immunotherapy (FZI), University Medical Center of the Johannes Gutenberg-University, 55131 Mainz, Germany

**Keywords:** TCF7L1, enteroendocrine cells, tuft cells, small intestine, WNT, Notch

## Abstract

Continuous and rapid renewal of the intestinal epithelium depends on intestinal stem cells (ISCs). A large repertoire of transcription factors mediates the correct maintenance and differentiation of ISCs along either absorptive or secretory lineages. In the present study, we addressed the role of TCF7L1, a negative regulator of WNT signalling, in embryonic and adult intestinal epithelium using conditional mouse mutants. We found that TCF7L1 prevents precocious differentiation of the embryonic intestinal epithelial progenitors towards enterocytes and ISCs. We show that *Tcf7l1* deficiency leads to upregulation of the Notch effector *Rbp-J,* resulting in a subsequent loss of embryonic secretory progenitors. In the adult small intestine, TCF7L1 is required for the differentiation of secretory epithelial progenitors along the tuft cell lineage. Furthermore, we show that *Tcf7l1* promotes the differentiation of enteroendocrine D- and L-cells in the anterior small intestine. We conclude that TCF7L1-mediated repression of both Notch and WNT pathways is essential for the correct differentiation of intestinal secretory progenitors.

## 1. Introduction

The gut epithelium plays a central role in immune surveillance, nutrient absorption, and synthesis of hormones. Its constant exposure to pathogens and xenobiotics leads to tissue injury. Abnormalities in the epithelial cell barrier and functions result in both acute disorders, such as colitis and diarrhoea, and chronic diseases, such as cancer. Therefore, gut epithelial cells must be both rapidly replaced and correctly specified. To maintain tissue homeostasis, intestinal stem cells (ISCs) constantly generate transit-amplifying progenitors that, in turn, are specified either along the absorptive enterocyte or the secretory lineages [1]. The secretory progenitors progressively differentiate towards mucous-secreting goblet, hormone-secreting enteroendocrine, immuno-modulating Paneth, and tuft cells.

WNT and Notch signalling molecules are crucial for stem cell self-renewal and balanced generation of absorptive and secretory cell types [1]. WNT signalling governs these processes by regulating a large number of genes encoding for transcription factors and components of signalling pathways [2]. On one side, cell cycle promoting genes, such as *c-Myc* and *cyclin D1* are transcriptionally activated by WNT signals [3]. On the other side, numerous transcription factors essential for the specification of undifferentiated progenitors along the secretory lineage (*Atoh1*), tuft (*Pou2f3*), and enteroendocrine cell lineages (*Neurog3*, *Neurod1*, *Foxa2,* and *Rfx3*) are direct targets of WNT signalling [2]. In the absence of WNT ligands, the downstream components of the WNT signalling T-cell-specific transcription factor 7 like 1 (TCF7L1) and TCF7L2, interact with corepressor proteins, such as Groucho/Transducin-like enhancer of split (GRO/TLE) [4] and C-Terminal-Binding Protein (CtBP) [5], that recruit histone deacetylases to the promoters of WNT target genes, leading to their silencing. Upon WNT stimulation, activated Dishevelled (DVL) promotes the dissociation of a large multiprotein complex containing β-catenin, Axin1/2, Adenomatous Polyposis Coli (APC), and two kinases: glycogen synthase kinase 3β (GSK3β) and casein kinase 1 (CK1) [6]. β-catenin then accumulates in the nucleus, where it associates with TCF7L2 and activates the transcription of WNT target genes [6]. In contrast, β-catenin binds TCF7L1 to remove it from DNA, which leads to TCF7L1 degradation [7]. Thus, TCF7L1 functions predominantly as a repressor of WNT-dependent genes [4].

Loss of *Tcf7l2* results in complete inhibition of cell proliferation in the neonatal and adult small intestine [8,9], indicating that this transcription factor is a non-redundant downstream effector of WNT signalling from the neonatal stage. However, TCF7L2 is not the only family member that is expressed in the gut epithelium. *Tcf7l1* is also expressed in the embryonic gut [10] and tuft cells [11]. A previous study showed that TCF7L1 is not required for either proliferation of the adult ISC or for differentiation along Paneth/goblet cell lineages [9]. However, the functions of TCF7L1 during embryonic gut development or differentiation of tuft and enteroendocrine cells were not examined.

In contrast to the adult intestinal epithelium, which is shaped in villi–crypt domains, the embryonic small intestine is a simple tube composed of a single epithelial layer surrounded by mesenchyme [12]. Between embryonic days 9.5 (E9.5) and E14.5, both the epithelial and mesenchymal cells proliferate at very high rates, which ensures a rapid expansion of the intestinal tube in length and circumference. Interestingly, the self-renewal of the embryonic intestinal epithelial progenitors is independent of WNT/β-catenin signalling. Consistently, WNT/β-catenin target genes, including *Axin2*, *Ascl2,* and *Lgr5* are expressed only in a few embryonic intestinal epithelial cells by E13.5 [10,13]. Around E14.5, the intestinal epithelium begins to differentiate, giving rise to enterocyte, goblet, and enteroendocrine cells [12].

To maintain ISCs and to specify ISC progenitors along either absorptive or secretory fates, the WNT/β-catenin pathway cooperates with the Notch pathway [14]. Interaction between the Notch receptor and delta-like (DLL) or jagged ligands leads to its cleavage [14]. The intracellular domain of the Notch receptor (NICD) then translocates to the nucleus and associates with recombination-signal-binding protein-J (RBP-J) transcription factor resulting in the activation of its target genes [15]. Loss-of-function mutation in *Rbp-j* causes rapid differentiation of transient-amplifying cells towards secretory goblet cells [16]. In contrast, ectopic activation of Notch signalling leads to the loss of secretory cells in the adult small intestine [17]. NICD/RBP-J blocks specification along the secretory lineage through the activation of *Hairy enhancer of split1/5* (*Hes1/5*) genes [15]. Moreover, *Hes1* transcription is also positively regulated by WNT/β-catenin signalling [18]. HES family members are transcription factors that in turn repress genes promoting the differentiation along secretory lineages, such as *Atoh1* [19]. Therefore, the Notch and WNT pathways converge on coregulated target gene promoters.

Here, we report that TCF7L1 deficiency results in the loss of secretory goblet and enteroendocrine cells during embryogenesis, and that these defects are associated with the upregulation of *Rbp-J* expression. We further demonstrate that TCF7L1 is required for the differentiation along tuft and enteroendocrine D- and L-cell lineages in the adult small intestine.

## 2. Materials and Methods

### 2.1. Mouse Strains

*Tcf7l1^tm1a(EUCOMM)Wtsi^* mice were obtained from *EMMA. Shh^EGFP-Cre^* mice were obtained from the Jackson Laboratory (Bar Harbor, ME, USA). CD1 mice were obtained from Charles River Laboratories (Cologne, Germany). *ROSA26::FLPe* mice were a gift from Thomas Hankeln, JGU, Mainz. Mouse colonies were maintained in a certified SPF animal facility per European guidelines. All mice were housed on a 12-h light/dark cycle with constant access to food and tap water. All the animal experiments were performed according to guidelines of the central animal facility institution (TARC, Mainz University Medical Center) representing those of the German Animal Welfare Act and the European Directive 2010/63/EU for the protection of animals used for scientific purposes. Breeding was approved by the local authorities (Kreisverwaltung Mainz-Bingen, Mainz). Reporting was carried out according to the ARRIVE guidelines for reporting in vivo experiments.

Mouse genotyping was performed using the following primers WT-for 5′-AGCAACCAAATGAAGGCTCAC-3′, WT-rev 5′-CTGCTGCCCCTCTTTTCATC-3′, mut-rev 5′-TCGTGGTATCGTTATGCGCC-3′, lox-for 5′-CTAACAAGTACAGAGGACTAACT-3′, and lox-rev 5′-ATCCGATATGCATACCACAAACT-3′. The expected sizes of wild-type and *LacZ* knock-in alleles were 340 bp and 200 bp, respectively. For floxed and wild type alleles, they were 250 bp and 200 bp, respectively. To detect Shh-Cre allele, CreF 5′-GGTGCGCTCCTGGACGTAGC-3′, and CreR 5′-GGGACAGCTCACAAGTCCTC-3′, primers were used, generating a 400 bp fragment.

### 2.2. Low Cell Number RNA-Sequencing

Small intestines were dissected from mouse embryos at day 13.5 (E13.5), cut into 2 mm pieces, and incubated for 10 min with 0.15 mg/mL collagenase (Sigma, Merck KGaA, Darmstadt, Germany) in PBS at 37 °C with shaking at 800 rpm. Single-cell suspensions were collected via centrifugation at 200× *g* for 5 min and resuspended in 200 mL of PBS supplemented with 2% goat serum. Cells were stained with APC-conjugated anti-EpCAM antibody 1:1000 (eBioscience, San Diego, CA, USA) for 30 min at room temperature. Living cells were gated via DAPI dye exclusion. Fluorescence-activated cell sorting analysis was performed using the BD FACS Aria II SORP cell sorter (85 μM nozzle).

For ultralow cell number RNA-sequencing five hundred EpCAM+ embryonic intestinal cells were isolated by FACS directly in 7 μL of lysis buffer (Takara Bio, Kusatsu City, Japan) supplemented with 5% RNase inhibitor and stored at −80 °C. cDNA was synthesised using SMARTer v4.0 kit (Takara Bio) according to the manufacturer’s instructions. Amplification was performed for 15 cycles. After cDNA fragmentation (Covaris, Woburn, MA, USA), libraries were prepared using the Ovation Ultralow v2 Library System (NuGEN, Houston, TX, USA) according to the manufacturer’s instructions.

### 2.3. RNA-Sequencing Data Analysis

Libraries were sequenced on an Illumina NextSeq 500 (NextSeq Control software v. 2.1.0.31) with a read length of 84 based and demultiplexed using bclfastq (v.2.19.1). Samples were mapped using STAR v2.5.2b [20] against iGenomes mm9. Reads per gene were counted using feature Counts v. 1.5.1 [21]. Read counts are based on uniquely mapping reads. For mapping as well as read counting the reference gene model downloaded from UCSC on 6 March 2013 was used (as provided with iGenomes). The differential expression analysis was performed using Bioconductor release 3.6 [22] and DESeq2 v1.18.1 [23] using standard parameters for testing and modelling as well as independent filtering. Finally, RPKM values were calculated per gene based on FPM (robust counts per million mapped fragments) values provided by DESeq2 and divided by the gene length and multiplied by 1000.

Sequencing-depth-normalised coverage tracks (bigwig) were generated using DeepTools v. 2.4.3 [24]. The raw sequencing data as well as the read counts per gene were deposited on the NCBI Gene Expression Omnibus database. The genes were considered significantly differentially expressed if they fit the following criteria: log_2_FC ≥ 0.5, FDR < 0.01, RPKM ≥ 50. Functional annotation analysis was performed using DAVID Functional Annotation Tool [25].

### 2.4. Quantitative PCR

For qPCR, 10 ng of cDNA generated with SMARTer v4.0 kit (Takara Bio) from EpCAM+ embryonic intestinal cells were used. Expression changes were normalized to *Tbp*. PCR primers were designed using Primer Blast “http://www.ncbi.nlm.nih.gov/tools/primer-blast/ (accessed on 17 May 2023)”. PCR was performed using SYBR green containing master mix kit (Applied Biosystems, Waltham, MA, USA) with ViiA™ 7 cycler (Applied Biosystems). A mean quantity was calculated from triplicate reactions for each sample.

### 2.5. RNA In Situ Hybridization

Embryos or adult small intestines were dissected and fixed in 4% PFA overnight. Prior to fixation, adult small intestines were divided into three equal parts—anterior, mid, and posterior. The following day tissues were dehydrated by incubation in a series of 30%, 50%, 70%, and 99% ethanol solutions and washed in xylol twice. Afterwards, tissues were embedded in paraffin and cut into 10 μm sections with the Leica RM2235 Rotary Microtome. Paraffin slides were incubated at 57 °C for 2 h, deparaffinized in xylol and rehydrated in a series of 99%, 70%, 50%, and 30% ethanol solutions, followed by PBS. Rehydrated sections were fixed in 4% PFA for 15 min, bleached in 6% H_2_O_2_ for 15 min, and treated with 10 ug/mL proteinase K solution for 10 min. The treated slides were immediately fixed in 4% PFA for another 15 min, washed twice in PBS, acetylated in freshly prepared 0.25% acetic anhydride in 100 mM Tris-Cl (pH 7.5) for 10 min. Then, tissues were equilibrated in 2 × SSC buffer, pH 5 and dehydrated in a series of 30%, 50%, 70%, and 99% ethanol solutions. The dehydrated slides were dried on air and hybridized with digoxygenin-labelled RNA probes for *Ghrl*, *Gip*, and *Sst* overnight at 63 °C. The following day, hybridized sections were washed in 5×, 2×, 1×, and 0.2 × SSC buffer at 60 °C and then proceeded to the blocking stage and were incubated overnight with sheep anti-digoxigenin antibody 1:3000 (Roche, Basel, Switzerland). The following day, slides were washed in TBSX buffer and stained with NBT/BCIP (Roche, Basel, Switzerland) until the signal developed. Afterwards, slides were dehydrated and mounted with ROTI^®^Histokitt (Carl Roth GmbH, Karlsruhe, Germany). Images were acquired on the Olympus IX2-UCB microscope.

RNA in situ hybridization for *Dclk1* was performed using RNAscope™ 2.5 HD Duplex Assay Kit (ACD Bio-Techne, Newark, CA, USA) following the manufacturer’s instructions. Target retrieval was performed for 15 min, and protease pretreatment was performed for 30 min. Images were acquired on the Olympus IX2-UCB microscope.

### 2.6. Immunohistochemistry and Detection of lacZ

Embryonic or adult small intestines were fixed for 16 h in 4% formaldehyde in PBS at 4 °C. Immunohistochemical analyses were performed on 10 μm paraffin-embedded gut tissue sections. Paraffin slides were incubated at 57 °C for 2 h, deparaffinized, and rehydrated. Antigen retrieval was performed by boiling in 40 mM sodium citrate buffer (pH 6.0) for 1 h. Endogenous peroxidases were blocked in 6% H_2_O_2_ for 10 min, washed twice in PBS and blocked with 5% goat serum and 0.1% NP-40. One hour later, the blocking solution was substituted by the primary antibody diluted in the blocking solution. The following antibodies and dilutions were applied: CHGA (1:500, ImmunoStar (Hudson, WI, USA)), 5-HT (1:500, Immunostar), GLP-1 (1:1000, Thermofisher (Waltham, MA, USA)), Ki67 (1:200, BioLegend (San Diego, CA, USA)), LYZ1 (1:500, Dako, Agilent, Santa Clara, CA, USA).

Slides were incubated with the primary antibody overnight, then washed thrice in PBS, 5 min each wash. Next, biotinylated secondary antibodies (Dianova, Biozol, Hamburg, Germany) were applied on the slides in 1:1000 dilution and incubated for two hours at room temperature. Next, the slides were washed thrice in PBS and incubated with Vectastain Elite ABC Reagent (Vector Laboratories) for 1 h at room temperature. The slides were washed in PBS and incubated with SIGMA FAST 3,3′-Diaminobenzidine Tablets (Sigma-Aldrich, St. Louis, MI, USA) dissolved in PBS. Afterwards, the slides were dehydrated and mounted with ROTI^®^Histokitt (Carl Roth GmbH, Karlsruhe, Germany). Images were acquired on the Olympus IX2-UCB microscope.

To detect LacZ-positive cells embryos were dissected in PBS, fixed with 0.2% glutaraldehyde for 20 min, washed 3 times with PBS/0.01% NP-40/0.01% Na-deoxycholate, and incubated for 16 h in staining solution (5 mM K_3_[Fe(CN)_6_], 5 mM K_4_[Fe(CN)_6_], 2 mM MgCl_2_, 0.01% NP-40, 0.01% Na-deoxycholate, and 1 mg/mL X-gal in PBS). Embryos were dehydrated through EtOH and xylol series and embedded in paraffin. Images were acquired with the Olympus IX2-UCB microscope.

### 2.7. Periodic Acid–SCHIFF (PAS) Staining

Paraffin sections (5 μm) were deparaffinized and hydrated to deionized water. Next, slides were immersed in periodic acid solution (Sigma-Aldrich, St. Louis, MI, USA) for 5 min at room temperature. Then, the slides were rinsed several times in distilled water and immersed in Schiff’s Reagent (Sigma-Aldrich, St. Louis, MI, USA) for 15 min. At the end of the incubation, slides were washed in running tap water for 5 min, dehydrated, mounted with ROTI^®^Histokitt (Carl Roth GmbH, Karlsruhe, Germany), and imaged on the Olympus IX2-UCB microscope.

### 2.8. Haematoxylin and Eosin (H&E) Co-Staining

Paraffin sections (5 μm) were deparaffinized, hydrated, and stained in Gill’s haematoxylin solution 2 (Sigma-Aldrich) for 3 min. After that, the slides were washed in running tap water for 5 min, rinsed in distilled water, and dipped in 95% ethanol. Next, the eosin co-staining was performed by immersing slides in the eosin solution (Sigma-Aldrich) for 1 min. Stained slides were washed in 95% ethanol, dehydrated, and mounted with ROTI^®^Histokitt (Carl Roth). Images were acquired on the Olympus IX2-UCB microscope.

### 2.9. Statistical Analysis

Information on sample size and statistical tests used for each experiment are indicated in the figure legends. All staining counts were analysed using a two-tailed nested *t*-test. Data are shown as means with SD, a *p*-value of ≤0.05 was considered significant. All data were tested for normality using D’Agostino and Pearson, Anderson–Darling, and Shapiro–Wilk tests. All described analyses were performed in GraphPad Prism v9.5.0.

## 3. Results

### 3.1. TCF7L1 Controls the Expression of Secretory Lineage Genes during Gut Development

To study the functions of TCF7L1 during gut development, we have used two mouse strains (Figure 1A). The first is a conditional *Tcf7l1-lacZ* line in which the splice acceptor-*lacZ* gene flanked by FRT sites is inserted in the fifth intron of the *Tcf7l1* gene [26]. Splicing of *lacZ* to the fifth exon of *Tcf7l1* generates both a null and a reporter allele. The second strain is a conditional *Tcf7l1* in which exon 6 is flanked by *loxP* sites, generating a frameshift and triggering nonsense-mediated decay of the mutant RNA upon Cre-mediated recombination [26]. We first examined the expression pattern of *Tcf7l1* at embryonic day 13.5 (E13.5), the earliest stage when stem cell, secretory and absorptive lineage specific genes are activated in the gut epithelium [10]. LacZ staining revealed that *Tcf7l1* is expressed in the epithelium of the anterior small intestine (Figure 1B). In contrast, the expression of *Tcf7l1-lacZ* reporter was not observed in the posterior half of the small intestine (Figure 1B), indicating that *Tcf7l1* is differentially expressed along the anterior–posterior axis at this developmental stage.

Mouse embryos lacking *Tcf7l1* die around E8.5 [27]. We therefore used the *Shh^Cre-EGFP^* allele, which displays Cre activity as early as E9.5 [28,29], to inactivate *Tcf7l1* in the developing gut epithelium. The small intestines of *Shh^Cre-EGFP^*:*Tcf7l1^lox/lox^* embryos were indistinguishable from wild type controls at E13.5. To explore whether the loss of *Tcf7l1* leads to ectopic activation of WNT target genes, we isolated EpCAM-positive intestinal epithelial cells using fluorescence-activated cell sorting (FACS) (Appendix A) from the mutant and control embryos and performed RNA-sequencing analysis.

We found that around 400 genes were upregulated and that around 800 genes were downregulated (log_2_FC ≥ 0.5, FDR < 0.01, RPKM ≥ 50) upon loss of *Tcf7l1* (Figure 1C–E, Appendix A). The adult stem cell signature genes *Olfm4*, *Hmgcs2,* and *Kcne3* were significantly upregulated, whereas *Lgr5* expression did not change compared to the wild type (Figure 1E). Of note, the expression of ISC signature genes varies along the anterior–posterior axis during embryonic development. While *Olfm4*, *Hmgcs2,* and *Kcne3* are expressed at higher levels in the anterior ISC progenitors, *Lgr5* is more highly expressed in the posterior ISCs progenitors [30]. Therefore, the changes in *Lgr5* expression in the anterior small intestine, where *Tcf7l1* is expressed, could be masked by the strong expression of the gene in the posterior epithelial cells. Moreover, the expression of enterocyte-specific genes, including *Fabp1*, *Fabp2*, *Lgals3*, *Aldob,* and *Ephx2,* were upregulated in *Tcf7l1* mutant compared to control cells (Figure 1C–E). Thus, TCF7L1 prevents precocious differentiation of the embryonic intestinal epithelium both towards ISCs and along the absorptive enterocyte lineage.

Functional annotation analysis of downregulated transcripts revealed enrichment for regulation of transcription (fold enrichment = 1.84, FDR = 5 × 10^−5^) and kinase activity (fold enrichment = 2.14, FDR = 1.1 × 10^−4^). Interestingly, we detected a strong decrease in the expression of genes encoding for transcription factors regulating the differentiation of enteroendocrine cells, including *Nkx6-2*, *Nkx6-3*, *Rfx6*, *Pdx1, and Foxa2* (Figure 1C–E and Appendix A), indicating that *Tcf7l1* is required for the differentiation of the embryonic epithelium along the enteroendocrine lineage. Consistently, we observed a complete loss of *ChgB* expression (Figure 1D), which is a marker of the enteroendocrine cells. Furthermore, markers of goblet cells, including *Fcgbp*, *Agr2,* and *Spdef* were also downregulated in *Tcf7l1* mutant embryos (Figure 1D,E). Our results show that *Tcf7l1* promotes differentiation of the intestinal epithelium towards both enteroendocrine and goblet cell lineages during embryogenesis.

### 3.2. TCF7L1 Is Necessary for Tuft Cell Differentiation

Single-cell RNA-sequencing analysis revealed that *Tcf7l1* is expressed in tuft cells in the adult small intestine [11]. RNA in situ hybridization analysis showed that *Tcf7l1* is expressed in a few spindle-shaped cells located mostly in the intestinal crypts (Figure 2A), suggesting that these cells could be tuft cell progenitors. To determine whether transcriptional changes during embryogenesis affect differentiation of the adult ISCs and to examine the functions of *Tcf7l1* in tuft cells, we analysed *Shh^Cre-EGFP^*:*Tcf7l1^lox/lox^* small intestines at the age of four months. Adult *Shh^Cre-EGFP^*:*Tcf7l1^lox/lox^* mice did not display any gross morphological or behavioural abnormalities. The villi–crypt architecture of mutant mice appeared normal (Appendix A).

Immunohistochemical staining for proliferation marker Ki67 was similar between mutant and wild type mice (Appendix A). Additionally, the numbers and localization of goblet (Figure 2B–D) and Paneth cells (Appendix A) were similar between *Shh^Cre-EGFP^*:*Tcf7l1^lox/lox^* and wild-type mice. In contrast, the number of tuft cells was significantly decreased in *Shh^Cre-EGFP^*:*Tcf7l1^lox/lox^* compared to wild-type mice as revealed by RNA in situ hybridization for *Doublecortin-like kinase 1* (*Dclk1)*, a marker of tuft cells (Figure 2E–G). The loss of tuft cells was observed in all parts of the small intestine along the anterior–posterior axis, with a 3-fold reduction in the anterior and posterior and a 3.5-fold in the middle parts (Figure 2G). Thus, we conclude that TCF7L1 is essential for the differentiation of tuft cells in the adult small intestine. However, transcriptional changes in either ISC signature genes or goblet progenitor markers caused by the loss of *Tcf7l1* during embryogenesis do not disturb stem cell proliferation and differentiation along the goblet/Paneth cell lineage in the adult gut, which confirms the results of the previous study [9].

### 3.3. TCF7L1 Is Dispensable for the Differentiation of Enterochromaffin Cells

Our transcriptome analysis of the intestinal epithelium revealed downregulation of *ChgB*, a marker of the enterochromaffin cells, and *Tox3* (Figure 1D and Appendix A) encoding for the HMG-box containing the transcription factor essential for differentiation along the enterochromaffin lineage [31] in *Shh^Cre-EGFP^*:*Tcf7l1^lox/lox^* compared to wild-type embryos. To determine whether the enterochromaffin cell population was affected by *Tcf7l1* knockout, we performed immunohistochemical analysis for the pan-marker of enteroendocrine cells (EECs), Chromogranin A (Figure 3A,B), and serotonin (5-HT), a marker for enterochromaffin cells (Figure 3D,E). Statistical analysis of the counts for CHGA-positive (Figure 3C) and 5-HT-positive (Figure 3F) cells did not show a significant difference between *Tcf7l1 KO* and wild-type mice, indicating that TCF7L1 is not required for the differentiation of enterochromaffin cells in the adult small intestine.

### 3.4. TCF7L1 Promotes the Differentiation of L- and D-Cells

The expression of Forkhead box A2 (*Foxa2*) was reduced eight-fold, and that of regulatory factor X 6 (*Rfx6*) was reduced three-fold in *Shh^Cre-EGFP^*:*Tcf7l1^lox/lox^* embryos compared to wild-type controls (Figure 1C–E and Appendix A). Winged helix/forkhead box transcription factor FOXA2 controls the differentiation of enteroendocrine progenitors along L- and D-cell lineages [32]. Moreover, *Rfx6* encoding for a winged helix transcription factor promotes the differentiation of all peptidergic EECs [33]. While the expression of both transcription factors is restricted to the enteroendocrine cells in the adult gut epithelium, they are broadly expressed in the embryonic epithelium [33,34]. The loss of *Foxa2* and *Rfx6* expression in *Tcf7l1* mutant embryos may change chromatin landscapes in the embryonic ISC progenitors leading to the alteration of adult ISC differentiation potential. Therefore, we examined the effect of TCF7L1 loss on the differentiation of L- and D- cells. Interestingly, we found a significant reduction (1.5-fold) of glucagon-like peptide 1 (GLP-1)-positive L-cells in the anterior small intestine of *Tcf7l1* knockout compared to wild-type mice. (Figure 4A–C).

The numbers of L-cells in the middle and posterior parts were also lower in *Tcf7l1KO* animals, yet a statistical significance was not reached (*p*-values for middle and posterior regions are 0.081 and 0.084, respectively). Furthermore, RNA in situ hybridization analysis for the D-cell marker *Somatostatin* (*Sst*) showed a significant reduction (1.5-fold) in D-cell number in the anterior part of the small intestine of *Tcf7l1* knockout mice compared to controls (Figure 5D,F). Altogether, these findings suggest that TCF7L1 promotes the differentiation of enteroendocrine cells along D- and L-cell lineages in the anterior small intestine.

### 3.5. Deletion of Tcf7l1 Does Not Affect the Differentiation of K- and X-Cells

RFX6 is also required for the differentiation of enteroendocrine progenitors along K- and X-cell lineages [33]. Therefore, we examined whether *Tcf7l1* loss in the intestinal epithelium resulted in similar defects. RNA in situ hybridization analysis for the expression of X-cell marker *Ghrelin* (*Ghrl*) did not reveal significant differences in the number of *Ghrl*-positive cells between *Shh^Cre-EGFP^*:*Tcf7l1^lox/lox^* and control small intestines (Figure 5A–C). Additionally, no significant differences were found in the number of *Gastric inhibitory peptide* (*Gip*)-positive K-cells between *Tcf7l1* knockout and wild-type mice (Figure 5D–F). These results indicate that TCF7L1 is not necessary for X- and K-cell differentiation.

## 4. Discussion

The WNT/β-catenin/TCF signalling pathway plays an important role in many developmental processes as well as during the maintenance and differentiation of adult stem cells. In this study, we examined the functions of TCF7L1 in the developing and adult gut epithelium. We show that in midgestation mouse embryos, TCF7L1 is required for the correct specification of intestinal epithelial cells along all secretory lineages. However, in the adult intestinal epithelium, TCF7L1 is necessary for the differentiation of secretory progenitors along the tuft cell lineage. Furthermore, a loss of *Tcf7l1* leads to a reduction in enteroendocrine GLP-1-secreting L-cells and somatostatin-positive D-cells in the anterior small intestine.

In the embryonic intestinal epithelium, *Tcf7l1* is expressed at least till E14.5 [13]. The decline of *Tcf7l1* transcription coincides with the activation of WNT/β-catenin signalling [10]. The repressor functions of TCF7L1 are essential for the self-renewal and differentiation of pluripotent stem cells and various tissue-specific progenitors. During early gastrulation, TCF7L1 represses pluripotency-associated genes, such as *Oct4*, *Sox2,* and *Nanog,* as well as genes promoting cell specification, including *FoxA2*, *Brachyury,* and *Lef1* [2,35]. This ensures a timely differentiation of the pluripotent epiblast cells toward the mesendoderm and neuroectoderm [27,35]. During neuronal differentiation, TCF7L1 inhibits the expression of transcription factor NEUROG1 and keeps neural progenitor cells in a self-renewing state to prepare them for further differentiation in the presence of WNT signals [36]. It is possible that TCF7L1 is required to repress genes promoting differentiation of the embryonic intestinal epithelial cells to secure an appropriate growth of the tissue. Consistent with the idea, we have observed that the expression of enterocyte-specific genes, including *Aldob*, *Alpi*, *Apoa4*, *Fabp1,* and *Fabp2,* was upregulated in *Tcf7l1* mutant embryos. These results indicate that TCF7L1 is required to lock the embryonic gut epithelium in an undifferentiated state. However, the expression of the WNT target genes, such as *Axin2*, *Lgr5,* and *Slc12a2,* was not affected. This confirms that the loss of TCF7L1 alone in the absence of WNT signals does not activate the transcription of WNT target genes.

Moreover, we found that the markers of goblet (*Agr2*, *Fcgbp,* and *Spdef*) and enteroendocrine cell progenitors (*Neurod1*, *Neurog3*, *Rfx6,* and *Foxa2*) were downregulated in the absence of *Tcf7l1*. These results suggest that TCF7L1 is also required either for the generation of secretory progenitors or for the expression of transcription factors promoting differentiation along goblet and enteroendocrine lineages. Previous studies showed that TCF7L1 binds enhancers of *Foxa2*, *Neurog3*, *Neurod1,* and *Spdef* and represses their transcription [2,37]. Therefore, TCF7L1 is required for differentiation along secretory lineages. A simultaneous increase in enterocyte markers suggests that TCF7L1 may regulate Notch signalling activity. Interestingly, we found that *Rbp-j* is significantly upregulated in *Tcf7l1* mutant epithelium (Appendix A). *Rbp-J* was also induced in *Tcf7l1* mutant embryonic stem cells [38]. In addition, hydroxymethylglutaryl-CoA synthase (HMGCS2), the expression of which is elevated in *Tcf7l1* mutants, promotes Notch signalling in the adult ISCs [39]. Finally, the activation of *Olfm4*, a target of Notch signalling [40], in *Tcf7l1* mutants indicates increased Notch activity. Thus, our data suggest that TCF7L1 might balance Notch signalling in the embryonic intestinal epithelium and that the changes in secretory lineage markers could be secondary to enhanced Notch signalling.

In the adult intestinal epithelium, TCF7L1 promotes differentiation along the tuft cell lineage but does not affect goblet or Paneth cell differentiation. Given that *Tcf7l1* is expressed mostly in tuft cells [11], this is an expected finding. Our data on tuft cells can be interpreted in two ways. First, *Pou2f3* encoding for the transcription factor essential for the differentiation of the secretory progenitors along the tuft cell lineage [41] is one of the TCF7L1 target genes in ESCs [2]. Therefore, TCF7L1 may be required for the transcriptional activation of *Pou3f2* in intestinal epithelial progenitors. This is unlikely, however, because TCF7L1 functions as a repressor. Second, the inhibition of WNT signalling leads to the increase in tuft cell numbers in the lung epithelium [42]. In contrast, upregulation of WNT signalling results in a loss of tuft cells. We propose that TCF7L1 promotes the differentiation of tuft cell progenitors by competing with TCF7L2 and reducing the transcriptional outcomes of WNT signalling.

Likewise, the inhibition of WNT signalling is crucial for the differentiation of enteroendocrine progenitors [43]. We found that the numbers of somatostatin-positive D-cells and GLP-1-positive L-cells were decreased in *Tcf7l1* mutant anterior small intestines compared to controls. It is possible that the anterior D- and L-cells are more sensitive to the levels of WNT signalling and thus require TCF7L1-mediated repression. Indeed, a previous study showed that cells located at the +4 position from the crypt bottom do co-express both WNT target genes *Lgr5 and Prom1* and enteroendocrine markers *Chga*, *Cck*, *Gip, and Ghrelin* [44], indicating that WNT signalling is active in enteroendocrine progenitors. However, we cannot exclude another possibility—that the action of TCF7L1 on the embryonic intestinal epithelium may change their chromatin landscapes, which in turn would affect later transcriptional programs in adult ISCs. Consistent with this model, we have observed that *Tcf7l1* is expressed in the anterior part of the small intestine during embryogenesis. As a result, the transcriptional changes that we detected were specific to the anterior intestinal epithelial cells. Further elucidation of transcriptional programs in *Tcf7l1* mutant adult ISCs at a single-cell level could validate this hypothesis.

## 5. Conclusions

Taken together, we have shown that TCF7L1 controls the differentiation of the embryonic gut epithelium. The discovery that TCF7L1 negatively regulates the expression of *Rbp-J* in intestinal epithelial cells provides an additional link between WNT and Notch pathways during embryogenesis, cell lineage specification, and neoplastic transformation. Finally, we propose that TCF7L1 acts to fine-tune levels of WNT-dependent transcription, which is essential for the differentiation of tuft cells in the adult gut.

## Figures and Tables

**Figure 1 cells-12-01452-f001:**
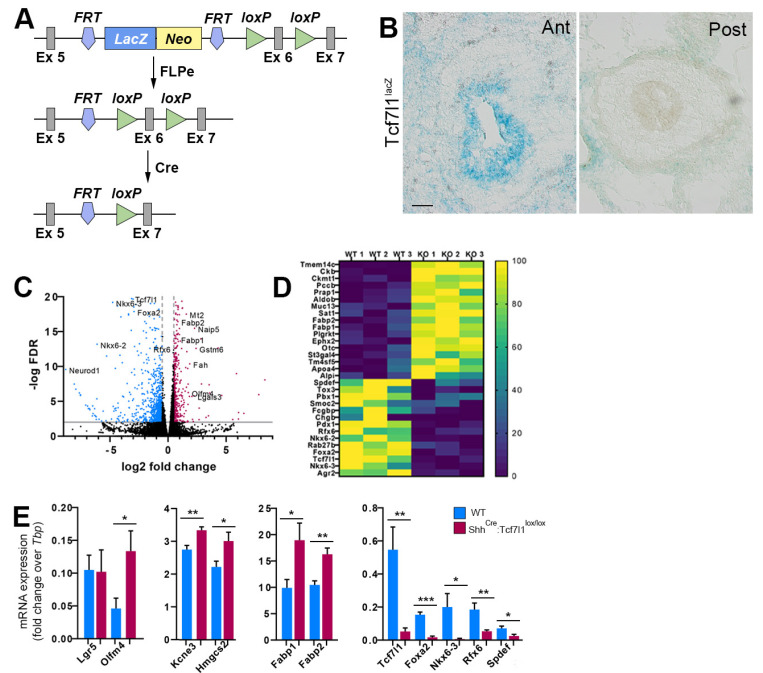
Transcriptional changes in the embryonic intestinal epithelium in the absence of *Tcf7l1*. (**A**) The strategy used to generate *Tcf7l1* conditional allele. Schematic representation of the “knockout-first” allele containing a splice acceptor: *lacZ* trapping cassette and a *neo* cassette inserted into the intron of a gene. After removal of the LacZ-neo cassette using *Rosa26^FLP^* recombinase, the “knockout-first” allele is converted to a conditional allele. Upon Cre mediated recombination the floxed exon 6 of the conditional allele is deleted, generating a frameshift mutation. (**B**) Transverse sections of the mouse small intestine from *Tcf7l1^lacZ^* embryos at E13.5 showing *Tcf7l1* expression as revealed by β-galactosidase activity. Ant stands for anterior, Post stands for posterior. Scale bar: 50 μm. (**C**) Volcano plot showing fold changes for genes differentially expressed between intestinal epithelial cells of *Tcf7l1* mutant and wild type embryos at E13.5. Genes that were upregulated in *Tcf7l1* mutants are shown in red. Genes that were downregulated in *Tcf7l1* mutants are depicted in blue. (**D**) Heatmap showing top up- and down-regulated genes in Tcf7l1 mutant epithelial cells compared to controls. The annotation bar indicates normalized expression for each gene. (**E**) Quantitative RT-PCR analyses measuring the levels of stem cell markers (*Lgr5*, *Olfm4*, *Kcne3* and *Hmgcs2*), enterocyte specific genes (*Fabp1* and *Fabp2*), enteroendocrine (*Foxa2*, *Nkx6-3*, *Rfx6*), and goblet cell (*Spdef*) markers in wild-type (blue) and *Shh^Cre-Egfp^:Tcf7l1^lox/lox^* (red) embryonic intestinal epithelium at E13.5. Values are normalized to the level of *Tbp*. Error bars are ±SD, n = 3 mice. * *p* < 0.05, ** *p* < 0.01, and *** *p* < 0.001 according to two-tailed Student’s *t*-test.

**Figure 2 cells-12-01452-f002:**
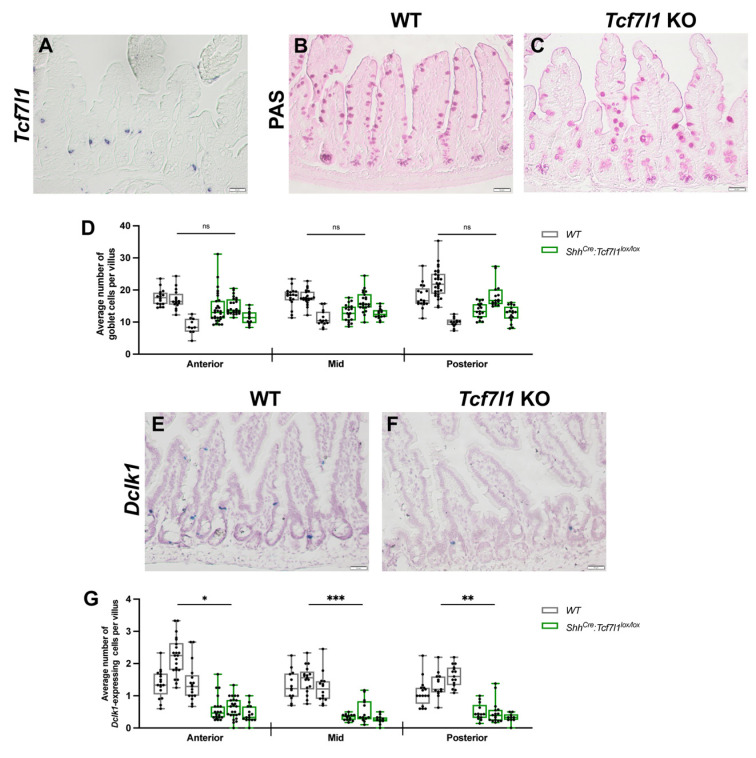
Loss of TCF7L1 function impairs the differentiation of tuft cells. (**A**) RNA in situ hybridization, showing *Tcf7l1*-expressing cells in mouse small intestinal epithelium. (**B**,**C**) Representative images of periodic acid–Schiff (PAS) staining of goblet cells in wild-type (**B**) and *Shh^Cre-EGFP^:Tcf7l1^lox/lox^* (**C**) mice. (**D**) Quantification of goblet cells in WT (grey) and *Shh^Cre-EGFP^:Tcf7l1^lox/lox^* (green) mice. Every dot shows an average number of stained cells per villus in a view field. (**E**,**F**) Representative images of *Dclk1* RNA in situ hybridization of the small intestinal epithelium of wild-type (**E**) and *Shh^Cre-EGFP^:Tcf7l1^lox/lox^* (**F**) mice. (**G**) Quantification of *Dclk1*-expressing cells in WT (grey) and *Shh^Cre-EGFP^:Tcf7l1^lox/lox^* (green) mice. Every dot shows an average number of stained cells per villus in a view field. Scale bar: 50 μm (**A**,**C**,**E**,**F**) Error bars are ±SD, n = 3 mice, ns stands for not significant, * *p* < 0.05, ** *p* < 0.01, and *** *p* < 0.001 according to a two-tailed nested *t*-test.

**Figure 3 cells-12-01452-f003:**
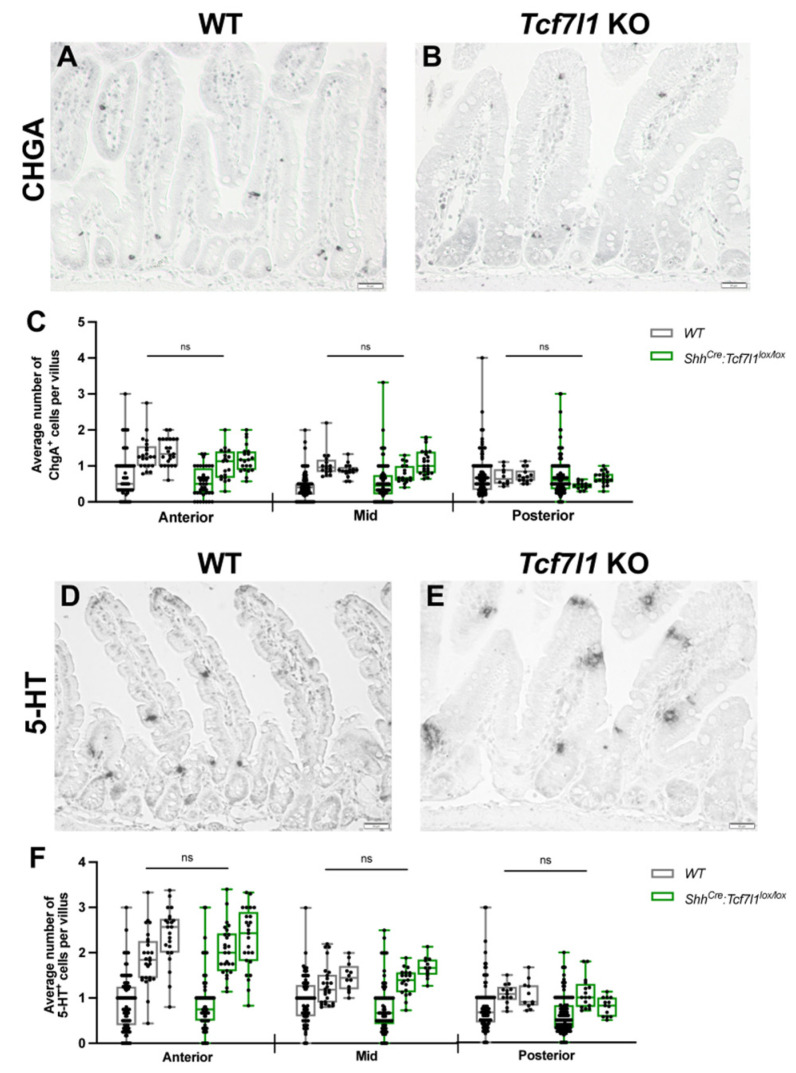
Tcf7l1 deletion does not affect enterochromaffin cell differentiation. (**A**,**B**) Representative images of immunostaining of CHGA-positive cells in the small intestinal epithelium of wild-type (**A**) and *Shh^Cre-EGFP^:Tcf7l1^lox/lox^* (**B**) mice. (**C**) Quantification of CHGA-positive cells in WT (grey) and *Shh^Cre-EGFP^:Tcf7l1^lox/lox^* (green) mice. Every dot shows an average number of stained cells per villus in a view field. (**D**,**E**) Representative 5-HT staining of wild-type (**D**) and *Shh^Cre-EGFP^:Tcf7l1^lox/lox^* (**E**) small intestine sections. (**F**) Quantification of the number of 5-HT-positive cells in WT (grey) and *Shh^Cre-EGFP^:Tcf7l1^lox/lox^* (green) mice. Every dot shows an average number of stained cells per villus in a view field. Scale bar: 50 μm (**A**,**B**,**D**,**E**). Error bars are ±SD, n = 3 mice according to a two-tailed nested *t*-test, ns stands for not significant.

**Figure 4 cells-12-01452-f004:**
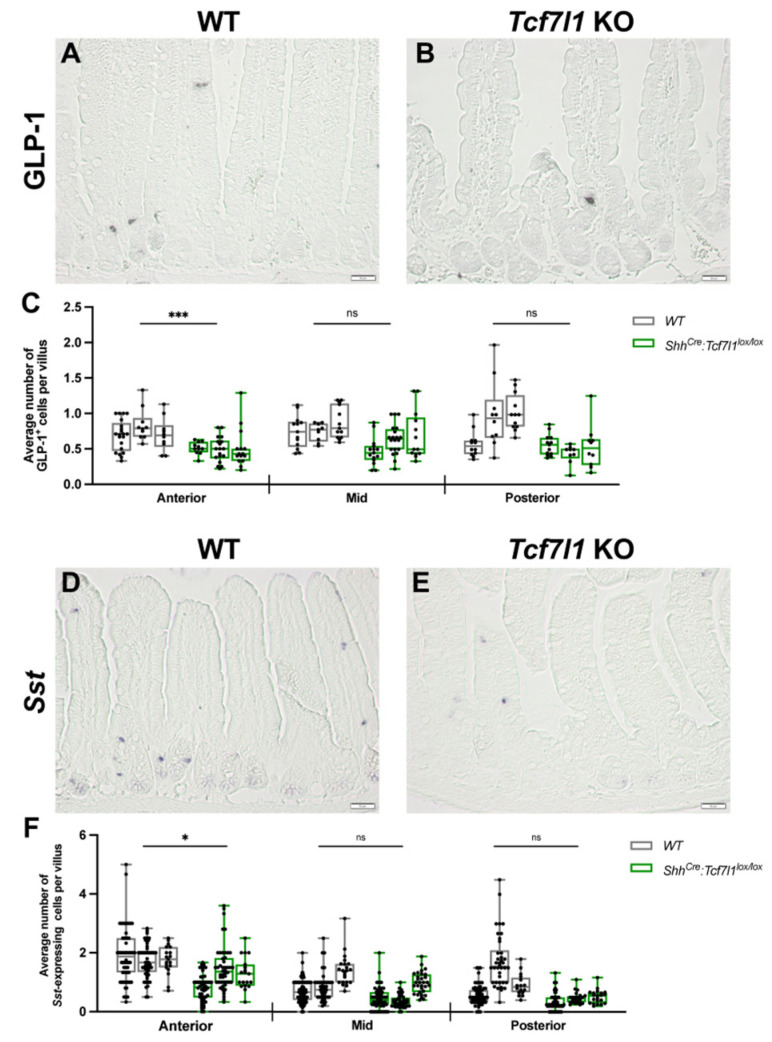
The numbers of L- and D-cells are decreased in *Tcf7l1* mutant small intestine. (**A**,**B**) Immunostainings for GLP-1 in wild-type (**A**) and *Shh^Cre-EGFP^:Tcf7l1^lox/lox^* (**B**) small intestine. (**C**) Quantification of GLP-1-positive cells in WT (grey) and *Shh^Cre-EGFP^:Tcf7l1^lox/lox^* (green) mice. Every dot shows an average number of stained cells per villus in a view field. (**D**,**E**) RNA in situ hybridization *for Sst* in wild-type (**D**) and *Shh^Cre-EGFP^:Tcf7l1^lox/lox^* (**E**) small intestines. (**F**) Quantification of *Sst*-positive cells in WT (grey) and *Shh^Cre-EGFP^:Tcf7l1^lox/lox^* (green) mice. Every dot shows an average number of stained cells per villus in a view field. Scale bar: 50 μm (**A**,**B**,**D**,**E**). Error bars are ±SD, n = 3 mice, ns stands for not significant, * *p* < 0.05 and *** *p* < 0.001 by two-tailed nested *t*-test.

**Figure 5 cells-12-01452-f005:**
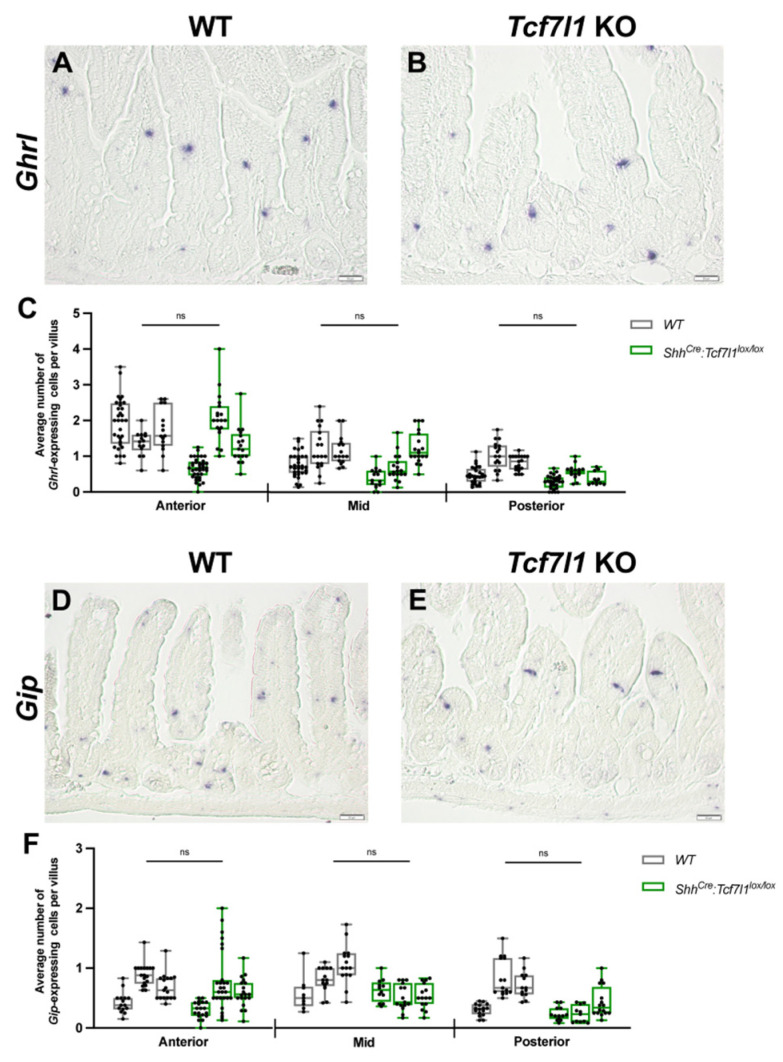
Effects of *Tcf7l1* ablation on the differentiation of X- and K-cells. (**A**,**B**) RNA in situ hybridization for *Ghrl* shows X-cells in the small intestinal epithelium of wild-type (**A**) and *Shh^Cre-EGFP^:Tcf7l1^lox/lox^* (**B**) mice. (**C**) Quantification of *Ghrl*-expressing cells in WT (grey) and *Shh^Cre-EGFP^:Tcf7l1^lox/lox^* (green) mice. Every dot shows an average number of stained cells per villus in a view field. (**D**,**E**) RNA in situ hybridization for *Gip* shows K-cells in the small intestinal epithelium of wild-type (**D**) and *Shh^Cre-EGFP^:Tcf7l1^lox/lox^* (**E**) mice. (**F**) Quantification of *Gip*-expressing cells in WT (grey) and *Shh^Cre-EGFP^:Tcf7l1^lox/lox^* (green) mice. Every dot shows an average number of stained cells per villus in a view field. Scale bar: 50 μm (**A**,**B**,**D**,**E**). Error bars are ±SD, n = 3 mice according to a two-tailed nested *t*-test, ns stands for not significant.

## Data Availability

The data discussed in this publication have been deposited in GEO database under accession number GSE226187.

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
