# Peer review of "TCF7L1 Controls the Differentiation of Tuft Cells in Mouse Small Intestine"

_cells, 2023, doi:10.3390/cells12111452_

Round 1
Reviewer 1 Report
In this manuscript the authors generated a gut specific conditional TCF7L1 knockout model. TCF7L1 was found to be expressed in the anterior portion of the small intestine and absent in more posterior regions. Deletion of TCF7L1 in the embryonic gut caused a up regulation of enterocyte markers and down-regulation of enteroendocrine and goblet cell lineages. In the adult intestine loss of Tcf7L1 resulted in impaired differentiation of tuft cells, as well as L- and D-cells.
This manuscript is scientifically sound and identifies novel functions for the TCF7L1 gene. This is interesting because a previous study had shown no obvious phenotypes associated with TCF7L1 knockout mice. Consequently, the paper will be of interest to researchers studying cell fate specification in the gut epithelium. I therefore recommend this paper for publication in Cells but the following point should be addressed first:
In the discussion the authors state: “Thus, we conclude that TCF7L1 is necessary to balance Notch signalling in the embryonic intestinal epithelium and that the changes in secretory lineage markers are secondary to enhanced Notch signalling.”
This statement is not convincingly supported by the evidence shown. The main issue is that no functional interrogation of Notch signaling was attempted in the manuscript. I would recommend the authors rewrite this passage to state that their data is “suggestive of” or that they hypothesize that changes in Notch signaling are required to mediate the cell fate changes observed.
Author Response
RESPONSE TO DECISION LETTER
We thank the editors and reviewers for their time, and for sharing their thoughts and expertise. We are glad that the referees found our work as interesting and important. We see their comments as constructive and aiming at improving our manuscript. In the revised version of the manuscript, we have addressed all concerns of the referees. We have also checked for typos. All changes are marked up using the “Track Changes” function. All references are relevant to the contents of the manuscript.
Point-to-point answers to the referees’ comments
Below are our answers to the referee’s comments. For the sake of clarity, the text from the referees is in italics, whereas our answers are in plain text.
Reviewer 1
We are glad the referee finds the manuscript scientifically sound and interesting.
However, the expert suggests further changes, which would improve the manuscript: “but the following point should be addressed first”
Reviewer 1 Comments for the Authors:
In the discussion the authors state: “Thus, we conclude that TCF7L1 is necessary to balance Notch signalling in the embryonic intestinal epithelium and that the changes in secretory lineage markers are secondary to enhanced Notch signalling.” This statement is not convincingly supported by the evidence shown. I would recommend the authors rewrite this passage to state that their data is “suggestive of” or that they hypothesize that changes in Notch signaling are required to mediate the cell fate changes observed.
We thank the expert for this suggestion. Following the recommendation of the referee, we have changed the text and now it reads: “Thus, our data suggest that TCF7L1 might balance Notch signalling in the embryonic intestinal epithelium and that the changes in secretory lineage markers could be secondary to enhanced Notch signalling.” (page 12, lane 436).
Reviewer 2 Report
Zinina et al investigate Tcf7l1 (Tcf3) functions in the small intestine in mice. The experiments are well designed and are executed well. Data presentation is very clear. Their finding is not necessarily novel or transformative, but is a great incremental addition of good knowledge to the field.
One major concern is the title of the manuscript. I do not find firm evidence that Tcf7l1 controls the differentiation of intestinal stem cells. Crucially, Tcf7l1 appears to be not expressed by adult intestinal stem cells. The authors need to revise the title so that it represents the findings in this work.
Minor concerns:
– Line 195, "0,2%" should be "0.2%".
– Line 329, what does "EECs" stand for?
– Line 412-415, citation 36 does not investigate NeuroD1 expression in neural progenitor cells. The authors need to carefully check their citations correctly reflect findings in the literature.
Author Response
RESPONSE TO DECISION LETTER
We thank the editors and reviewers for their time, and for sharing their thoughts and expertise. We are glad that the referees found our work as interesting and important. We see their comments as constructive and aiming at improving our manuscript. In the revised version of the manuscript, we have addressed all concerns of the referees. We have also checked for typos. All changes are marked up using the “Track Changes” function. All references are relevant to the contents of the manuscript.
Point-to-point answers to the referees’ comments
Below are our answers to the referee’s comments. For the sake of clarity, the text from the referees is in italics, whereas our answers are in plain text.
Reviewer 2
We are glad that the expert finds the manuscript as clearly written, “experiments designed and executed well, and is a great incremental addition of good knowledge to the field”
However, the expert pointed out also weaknesses, which were addressed in the revised version of the manuscript.
Reviewer 2 Comments for the Authors:
One major concern is the title of the manuscript. I do not find firm evidence that Tcf7l1 controls the differentiation of intestinal stem cells. Crucially, Tcf7l1 appears to be not expressed by adult intestinal stem cells. The authors need to revise the title so that it represents the findings in this work.
We agree with the expert. A new title is: “TCF7L1 controls the differentiation of tuft cells in mouse small intestine”.
Minor concerns:
– Line 195, "0,2%" should be "0.2%".
We thank the referee for this comment. The text is changed accordingly.
– Line 329, what does "EECs" stand for?
We thank the referee for this comment. The abbreviation EECs is now explained as enteroendocrine cells (EECs) (page 8, lane 329).
– Line 412-415, citation 36 does not investigate NeuroD1 expression in neural progenitor cells. The authors need to carefully check their citations correctly reflect findings in the literature.
The expert is right. While the binding of TCF7L1 to Neurod1 promoter in embryonic stem cells (Cole, M.F.; Johnston.; S.E.; Newman, J.J; Kagey, M.H.; Young, R.A. Tcf3 is an integral component of the core regulatory circuitry of embryonic stem cells. Genes Dev. 2008, 22, 746-755. doi: 10.1101/gad.1642408) and the requirement of WNT signalling for the activation of Neurod1 transcription upon neural progenitor cell (NPC) differentiation were shown (Kuwabara T, Hsieh J, Muotri A, Yeo G, Warashina M, Lie DC, Moore L, Nakashima K, Asashima M, Gage FH. Wnt-mediated activation of NeuroD1 and retro-elements during adult neurogenesis. Nat Neurosci. 2009 Sep;12(9):1097-105. doi: 10.1038/nn.2360), the presence of TCF7L1 on the Neurod1 promoter in NPCs was not demonstrated. Also, whether TCF7L1 represses Neurod1 expression in NPCs is an open question. Therefore, we have edited this sentence and now it reads as: “During neuronal differentiation, TCF7L1 inhibits the expression of transcription factor NEUROG1 and keeps neural progenitor cells in a self-renewing state to prepare them for further differentiation in the presence of WNT signals [36].”